

# A graph neural network approach to basin-scale river network learning: The role of physics-based connectivity and data fusion

Alexander Y. Sun[1], Peishi Jiang[2], Zong-Liang Yang[3], Yangxinyu Xie[4], and Xingyuan Chen[2]

[1]Bureau of Economic Geology, The University of Texas at Austin, Austin, TX, USA
[2]Pacific Northwest National Laboratory, Richland, WA, USA
[3]Department of Geological Sciences,The University of Texas at Austin, Austin, TX, USA
[4]Department of Computer Science, The University of Texas at Austin,Austin, TX, USA

**Correspondence:** Alexander Y. Sun (alex.sun@beg.utexas.edu)

**Abstract.** Rivers and river habitats around the world are under sustained pressure from anthropogenic activities and the changing global environment. Our ability to quantify and manage the river states in a timely manner is critical for protecting the public safety and natural resources. Vector-based river network models have enabled modeling of large river basins at increasingly fine resolutions, but are computationally demanding. This work presents a multistage, physics-guided, graph neural network (GNNs) approach for basin-scale river network learning and stream forecasting. GNN models are pretrained using a high-resolution vector-based river network model, and then fine-tuned with in situ streamflow observations, after which a post-processing data fusion step is proposed to propagate residuals over the entire network to correct predictions. The GNN-based framework is demonstrated over a snow-dominated watershed in the western U.S. consisting of 552 reaches. A series of experiments are performed to test different training and imputation strategies. Results show the trained GNN model can effectively serve as a surrogate model of the process-based model with high accuracy, with median Kling–Gupta efficiency (KGE) greater than 0.97. Application of the graph-based data fusion further reduces mismatch between the GNN model and observations, with as much as 50 percent KGE improvement over some cross-validation gages. Additionally we exploit and demonstrate a graph coarsening procedure that achieves comparable predicting skills at only a fraction of training cost, thus providing important insights on the degree of physical realism needed for developing large-scale GNN-based river network models.

## 1 Introduction

Rivers play a critical role in the hydrosphere, enabling and regulating hydrological, geomorphic and ecological processes along and adjacent to the riverine environment (De Groot et al., 2002; Dai and Trenberth, 2002). Rivers



around the world are also under sustained pressure from human activities, losing free-flowing connectivity over time due to the construction of dams, levees, and other hydroinfrastructures for providing societal goods and services (e.g., hydropower generation) (Best, 2019). About half of the global river reaches now show signs of diminished connectivity (Grill et al., 2019). The changing climate and population growth further exacerbate the existing stress

on river systems by modulating the spatial and temporal patterns of floods and droughts, including their frequency, magnitude and timing (Winsemius et al., 2016; Blöschl et al., 2017; Dottori et al., 2018), and by reducing the rivers' natural ability to absorb disturbances and buffer the ecosystem (Palmer et al., 2008). Thus, our ability to quantify and manage the river states and fluxes in a timely manner has become more important than ever for protecting the public safety and adapting to the changing environment. Stream gauges can provide direct measures of river

discharges, but the utility of which is hindered by the poor coverage of gauge networks in many parts of the world. Instead, hydrological models are broadly used to predict river discharges at ungauged locations (Hrachowitz et al., 2013; Beck et al., 2015).

The fidelity of a hydrological model hinges on a number of factors, including the accuracy of forcing data, soundness of process parameterization, and the realism of river network geometry used in the model (Bierkens et al., 2015).

In many distributed and semi-distributed models, river networks are delineated on the same grid as the underlying hydrological models or land surface models (Döll et al., 2003; Van Beek et al., 2011; Pokhrel et al., 2012; Alfieri et al., 2013). Prescribing river networks over coarse resolution grids may introduce misrepresentation in river routing models, leading to inaccurate river discharge estimates and misalignment with local application needs (Zhao et al., 2017; Mizukami et al., 2021). While representation of river networks can always be improved by refining the grid

resolution, in reality such effort is often constrained by computational resources, especially for large scale simulations (Yamazaki et al., 2013; Bierkens et al., 2015).

In the past decade, vector hydrography has (re)emerged as an alternative to the conventional grid-based approaches for large river basin simulations (David et al., 2011; Lehner and Grill, 2013; Yamazaki et al., 2013; Lin et al., 2018). In a vector network representation, the land surface of a river basin is discretized into unit catchments (polygons)

and the river reaches connecting them by using a high-resolution digital elevation model (Saunders, 2000). Unlike the grid-based representation, in a vector-based river model the unit catchments serve as calculation units, and the time evolution of state variables is solved by calculating the flux exchange between each unit catchment and the next downstream unit catchment along a prescribed river network (Yamazaki et al., 2013). Direct benefits of such a discretization scheme are (a) river geometries are realistically represented; (b) river reach distances are relatively

evenly distributed, allowing for greater computing time steps and thus gaining more computational efficiency; (c) the unit catchment resolution can be easily improved by applying more detailed polygon outlines; and (d) hydrologic features such as lakes and irrigation lines can be modeled as additional vector features attached to the catchments and river reaches (Mizukami et al., 2021).

The vector-based river representation is behind the National Water Model (NWM), which operationally generates

short- and long-range river discharge forecasts for more than 2.7 million reaches in the U.S. (Lin et al., 2018;





Salas et al., 2018). Vector-based river network simulations were also used to create validation datasets for the upcoming Surface Water and Ocean Topography (SWOT) satellite mission, designed to provide global observations of changing water levels in large rivers, lakes, and floodplains (Biancamaria et al., 2016; Lin et al., 2019). Despite their improved efficiency, vector-based river routing models are still computationally demanding, requiring special
domain decomposition techniques for parallel computation (David et al., 2011; Mizukami et al., 2021)

The renewed interests in vector hydrography have been driven by an increased demand for hyperresolution terrestrial hydrology models on the one hand (Bierkens et al., 2015), and the deluge of high-resolution Earth observation data on the other (Sun and Scanlon, 2019; Reichstein et al., 2019). Ultimately, the Earth science community envisions the development of Earth system digital twins, aiming to provide digital replica of the real world through
high-fidelity simulations and observations (Bauer et al., 2021). Toward that vision, two active veins of research are currently underway. One is continuously improving the physical realism of process representation within the current Earth system models at all scales and across all subsystem interfaces, which is a daunting task even with today's extreme-scale, high-performance computing power (Schulthess et al., 2018). The other is augmenting process-based models with artificial intelligence/machine learning (AI/ML) techniques, which has attracted significant attentions
in the past several years (Shen, 2018; Sun and Scanlon, 2019; Reichstein et al., 2019; Camps-Valls et al., 2021; Kashinath et al., 2021; Pathak et al., 2022). Existing AI/ML works related to hydroclimate modeling may be classified as (a) data-driven techniques (Kratzert et al., 2019b; Le et al., 2021; Sun et al., 2021; Feng et al., 2021); (b) hybrid process-based/ML models (Rasp et al., 2018; Yuval and O'Gorman, 2020; Nonnenmacher and Greenberg, 2021); and (c) physics-guided, post-processing (PGPP) techniques (Ham et al., 2019; Sun et al., 2019; Yang et al.,
2019; Feng et al., 2020; Willard et al., 2020; Lu et al., 2021; Kashinath et al., 2021; Pathak et al., 2022). Although in the literature the latter two categories are sometimes combined, in the context of this discussion the former category is seen as providing ML-based parameterization schemes (e.g., subgrid processes) within a process-based model (Rasp et al., 2018; Reichstein et al., 2019), while the latter category mainly leverages outputs of process-based models and first-order physics principles in an ad hoc manner. ML-based PGPP methods can provide added values
to existing process models, such as improved predictability, reduced bias, and/or computational efficiency, without requiring significant modifications to the existing scientific computing workflows and codes. This largely explains the popularity of PGPP paradigm in the Earth science community.

So far, only a few studies have exploited the use of ML in vector hydrography. This is partly because many of the deep learning techniques in use today are originated from the computer vision and natural language processing
literature, dealing mostly with gridded or sequence data. On the other hand, many types of data in natural and social sciences, such as weather stations, river networks, and social networks, are characterized by graph-like data structures (i.e., nodes and node links) that do not conform to the Euclidean geometry. Various graph neural network (GNN) models have been developed (Bruna et al., 2013; Kipf and Welling, 2016; Bronstein et al., 2017; Zhou et al., 2018) to learn the graph structured data. Like their counterpart for learning image-like data (e.g., convolutional
neural networks or CNNs), GNNs are designed to extract high-level features from input data through the the so-



called neural message passing and aggregation process, consisting of a series of algebraic operations to progressively encode nodes' features and their local structures (e.g., the number of neighbors) as latent representations (i.e., low-dimensional embeddings) (Kipf and Welling, 2016; Hamilton et al., 2017). Parameters of the embeddings are then learned from the training data through back propagation. GNNs, when combined with temporal learning algorithms

(e.g., recurrent neural network or RNN), comprise powerful tools for disentangling highly complex, spatial and temporal relationships. We point out that the graph theory, which is at the foundation of GNNs, has been widely applied in geosciences to model geological processes, for example, formation of river deltas (Phillips et al., 2015; Tejedor et al., 2018). However, GNNs focus more on learning the latent-space representations for downstream tasks (e.g., classification and regression) than on abstracting graph-level statistics, as commonly done under most graph

theoretic studies (Hamilton et al., 2017).

Jia et al. (2021) used a physics-guided, recurrent graph convolution network (GCN) model to predict streamflow and water temperature in a catchment of the Delaware River basin in the U.S. In their work, physics constraints are imposed in multiple ways: the actual river reach lengths and upstream/downstream connections are used to construct a weighted adjacency matrix; a process-based streamflow and water temperature model is used to generate synthetic

training samples for ungauged reaches; and finally data collected from gauged locations are used in an extra loss term to enforce physical consistency with observations. The authors reported their recurrent GNN model generally gives better performance than a baseline RNN model, but may yield large errors at unobserved river reaches and reaches with extremely low flows. Sun et al. (2021) adapted and compared the performance of several recurrent GNN architectures for predicting streamflows of basins in the Catchment Attributes and Meteorology for Large-Sample

Studies (CAMELS) dataset, which includes meteorological forcing, basin static attributes, and observed streamflow time series for 671 basins in the conterminous U.S. (Newman et al., 2015). The GNN algorithms they investigated include several basic GNNs, such as the GCN (Kipf and Welling, 2016) and ChebNet (Defferrard et al., 2016), as well as a more complex GNN model, the GraphWavenet (GWN for short) (Wu et al., 2019). They used hydrosimilarity as a distance measure to connect the spatially scattered basins, treating each basin and its static attributes as node

and node features. Their results show the GWN gives the best overall performance, while models based on the basic GNN layers perform worse than a baseline model trained using the long short-term memory (LSTM) network (Hochreiter and Schmidhuber, 1997). Chen et al. (2021) adopted a heterogeneous recurrent graph model to predict stream temperatures, in which river reaches and dams are represented as separate graphs. For each river reach the authors used gating variables to control the information flow from upstream river reaches and reservoirs, in

addition to the antecedent states of the river reach itself. Similar to (Jia et al., 2021), observations are incorporated directly in an extra loss term during training, as well as for adjusting the hidden states. These recent studies have demonstrated the potential of GNNs for vector-based river modeling. Specifically, GNNs may allow fine-grained control of information exchanges at the node and edge levels for incorporating the physical realism, an aspect that is missing in many other ML algorithms commonly used to model spatiotemporal datasets (e.g., random forests and





RNNs). In the remainder of this discussion, we shall use node and reach interchangeably when streamflow in the reach is implied.

Despite their potential promise, remaining questions pertaining to GNN applications in the vector hydrography are (a) the degree of physical realism that is needed for a GNN model to learn river network representations; (b) the generalization and scalability of GNN models, and (c) data fusion. To understand these research needs,

we highlight the main differences between river networks and many other types of networks. River networks are hierarchical, with downstream discharges reflecting the integrated hydrologic contributions from all upstream reaches and catchments (Weiler et al., 2003). Reaches adjacent to each other tend to share similar runoff generation processes because of similarities in catchment physiographic properties and meteorological forcing. These unique aspects of river networks imply that information passing in a river network should be multi-directional, rather than strictly

following the network topography as prescribed through the physics-based connectivity. Identifying and modeling the multi-directional and heterogeneous information transfer mechanisms using GNNs is an active research topic (Zhou et al., 2018). Perhaps a more practical question is related to graph generalization and scalability. The river networks considered in Jia et al. (2021) and Chen et al. (2021) include 42 and 56 river reaches, respectively, which are relatively small. In comparison, a typical basin at the U.S. Geological Survey's (USGS) 8-digit hydrological unit code

(HUC-8) level contains $\mathcal{O}(10^2 - 10^3)$ river reaches, and at the largest HUC-2 regional basin level, the river network may include $\mathcal{O}(10^4 - 10^5)$ reaches (Simley and Carswell Jr, 2009). State-of-the-art GNN models have demonstrated capabilities to handle large scale graphs containing $\mathcal{O}(10^8)$ nodes on classification problems (Hu et al., 2020; Wang et al., 2021). Learning large-scale, vector river networks, however, is still a challenging topic because of the dynamic nature of the problem and computing memory requirements. A fruitful research direction may be exploring the

graph sparsification or coarsening in a way that preserves the spatiotemporal structure of the river system. Finally, data fusion on graph-based, river network models has not been systematically studied, but represents an important class of post-processing techniques for watershed modeling. In general, post-processing techniques, as their names suggest, attempt to refine model outputs using new observations obtained after model simulations. In streamflow forecasting, post-processors have been used to correct the biases and dispersion errors in raw forecasts, downscale

the forecasts to the scale of applications, and generate forecast ensembles that preserve the spatiotemporal structure of river discharges (Todini, 2008; Weerts et al., 2011; Li et al., 2017). Unlike data assimilation, data fusion as defined and used in the context of this work is generally decoupled from the original model.

In light of the aforementioned challenges, this study was conducted with the following two objectives in mind, namely, (a) evaluate the role of physics-based connectivity in GNN river network surrogate modeling, and (b) adapt

and investigate the efficacy of a graph-based data fusion technique. Main contributions of this work are we have developed a methodology consisting of pretraining, fine-tuning, and data fusion steps to significantly improve the performance of GNN models; we show that the degree of realism required for a GNN surrogate model to catch spatiotemporal basin flow dynamics largely depends on the parameter structure of the underlying physics-based model. The remainder of this paper is organized as follows. In Section 2, we describe data and data processing techniques





used in this study. Section 3 focuses on the theoretical background of GNN and data fusion algorithms. Section 4 describes the demonstration study area and experimental design, which are followed by results and discussion in Section 5 and conclusions in the last section.

## 2 Data and Data Processing

### 2.1 National Water Model

NWM is a continental-scale, distributed, hydrological modeling framework implemented and operated by the U.S. National Weather Service for providing short-range (18 hour), medium-range (10 days) and long-range (30 days) streamflow forecasts in the U.S. (Cosgrove et al., 2016). It is based on the WRF-Hydro community model, which is both a standalone model and a coupling architecture to facilitate the exchange between the Weather Research and Forecasting (WRF) atmospheric model and components of a land surface model (e.g., surface runoff, channel flow,

lake/reservoir flow, subsurface flow, and land-atmosphere exchanges) (Gochis et al., 2018). WRF-Hydro supports surface runoff routing over vector-based river networks. The network topology used in NWM is derived from the U.S. National Hydrography Dataset Plus (NHDPlus), a georeferenced, hydrologic dataset incorporating 1:100,000-scale national stream network and a 30-m national digital elevation dataset, in addition to a large number of river and catchment attributes for enhancing network analyses (McKay et al., 2012; Moore and Dewald, 2016). We

primarily used the NWM v2.0 retrospective simulation data that was available when this study was initiated. NWM v2.0 contains NWM retrospective simulation outputs in hourly time steps at 2,729,076 NHDPlus reaches for the period 1993/01/01–2018/12/31. As part of the sensitivity analysis, we also tested our approaches on NWM v2.1 retrospective data, which covers a longer 42-year period (1979/02/01–2020/12/31). A subset of NWM parameters are calibrated by model developers using historical streamflow data at limited basins, but no nudging is applied on

the retrospective runs (Cosgrove et al., 2016). NWM v2.0 streamflow data is downloaded from a data server hosted by Hydroshare (Johnson and Blodgett, 2020), while NWM 2.1 streamflow data is downloaded from National Oceanic and Atmospheric Administration's (NOAA) Amazon Web Services data repository (NOAA, 2022). Python scripts are used to automate the remote subsetting and downloading of all NWM streamflow data for any U.S. basin of interest. All data is aggregated into daily steps.

### 2.2 Meteorological forcing and streamflow

NWM v2.0 is driven by forcing data resampled from the North American Land Data Assimilation System (NLDAS), which is originally available on a 1/8 degree grid (~14km at the equator) (Xia et al., 2012), while NWM v2.1 is driven by the 1-km Analysis of Record for Calibration (AORC) dataset that is not publicly available at the time of this writing (Kitzmiller et al., 2018). In this study, we used Daymet, which provides gauge-based, gridded estimates of

daily weather and climatology variables over the continental North America, including daily minimum and maximum



temperature, precipitation, vapor pressure, shortwave radiation, snow water equivalent (Thornton et al., 2012). The spatial resolution of Daymet is 1 km × 1 km and the temporal coverage is from 1950 through the end of the most recent full calendar year. Although built upon similar gauge data, Daymet data is likely different from the meteorological forcing data used in NWM because of the different interpolation and extrapolation schemes used to create it. Combining Daymet with antecedent NWM outputs as predictors may thus indirectly achieve the effect of using multiple forcing data, which has been shown to improve the generalization skill of Earth science ML models (Sun et al., 2019; Kratzert et al., 2021). Daymet data is downloaded by programmatically calling the Daymet web services (ORNL, 2022) and getting data closest to the centroid of each reach in a river network. Streamflow gage data is downloaded from USGS' National Water Information System by using the USGS Python package for water data retrieval (USGS, 2022a).

## 2.3 River network construction

River network for a basin under study may be extracted from the NHDPlus database by performing the following steps. First, the NHDPlus v2.1 geodatabase covering the basin is downloaded from the U.S. Environmental Protection Agency (EPA) data server (EPA, 2022). The basin mask is used to crop the NHDFlow shape layer included in the NHDPlus geodatabase, which is then joined with the PlusFlow table, also from NHDPlus. After this step, we have all reach attributes, including identification number of each river reach (referred to as COMID in NHDPlus), the upstream/downstream reaches of each reach, the reach type (e.g., stream or artificial path) and reach length, that are necessary for building a river network and populating the node features. We used a Python script to recursively traverse all river reaches to gather reach attributes and build the network (i.e., in terms of graph adjacency matrix). The reach COMIDs corresponding to USGS gauge locations are also obtained for mapping purposes. All watershed boundaries used in this study are extracted from the Watershed Boundary Dataset, which includes basin boundaries at various HUC levels (USGS, 2022b). For the graph imputation/interpolation demonstration, we used the pour points corresponding to the HUC-12 basins (Price, 2022).

## 3 Methodology

We start by introducing some notations. A graph is represented by $\mathcal{G}(\mathcal{V}, \mathcal{E})$, where $\mathcal{V} = \{v_i\}_{i=1}^{N}$ is a set of $N$ nodes and $\mathcal{E} = \{e_{ij}\}$ is a set of edges connecting node pairs $(v_i, v_j)$ for $v_i \in \mathcal{V}$ and $v_j \in \mathcal{V}$. The neighborhood of a node is a subset of nodes connected to it, $\mathcal{N}(v) = \{u \in \mathcal{V} \mid (u, v) \in \mathcal{E}\}$. Node connections are specified by the adjacency matrix $\mathbf{A} \in \mathbb{R}^{N \times N}$, of which an element $a_{ij}$ is equal to 1 if nodes $v_i$ and $v_j$ are connected and 0 otherwise. A graph can be either undirected (edge is bidirectional) or directed (edge direction matters). The adjacency matrix may also be weighted, in which case elements of $\mathbf{A}$ would be decimal numbers describing the affinity or similarity between two nodes. The graph feature matrix is denoted as $\mathbf{X} \in \mathbb{R}^{N \times D}$, with its rows representing node feature vectors





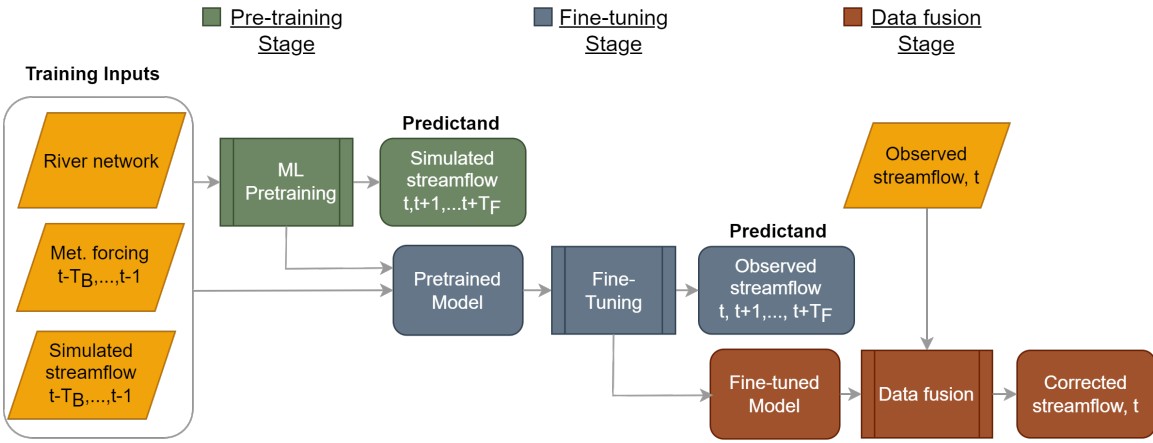

**Figure 1.** An ML-based workflow for basin-scale streamflow forecasting. The workflow consists of three stages, pretraining, fine-tuning, and data fusion. During pretraining, an ML model is trained to learn the input-output mapping as governed by the physics-based National Water Model. Inputs include river network attributes and connectivity, meteorological forcing, and antecedent simulated streamflow. The predictand is simulated streamflow, where $T_B$ and $T_F$ denote the lengths of lookback and forecast periods. In the fine-tuning step, the ML model is trained to minimize mismatch with historical streamflow observations. During data fusion, ML streamflow predictions are adjusted through graph-based residual propagation. Both pretraining and fine-tuning are performed offline, but data fusion can be done in real time.

$\mathbf{x}_i \in \mathbb{R}^D, i = 1, \ldots, N$. In the dynamic setting considered in this work, the node features vary with time and the graph feature matrix is denoted by $\mathbf{X}^t$.

To develop a GNN-based, end-to-end framework for vector-based river network modeling, we propose a three-
235 stage workflow as shown in Fig. 1. In Stage I, a GNN-based model is trained through supervised learning by using the meteorological forcing data and NWM outputs. This pretraining step is essentially a surrogate modeling process that learns the spatiotemporal rainfall-runoff patterns governed by NWM and meteorological forcing. The predictors include the six meteorological forcing variables from Daymet and the NWM simulated streamflow for a lookback period of $T_B$, as well as the adjacency matrix $\mathbf{A}$ describing the river network connectivity. A neural network is
trained to approximate the mapping $\mathcal{F}(\mathcal{X}^t, \mathbf{y}^t)$, where $\mathcal{X}^t$ is a collation of predictors $\{\mathbf{X}^i\}_{i=t-1}^{t-T_B}$, $\mathbf{y}^t \in \mathbb{R}^N$ is the NWM outputs for the entire river network, and the forecast can be done for $T_F$ steps in the future. In Stage II, a fine-tuning step is used to correct the pretrained GNN model by utilizing historical streamflow data available in the training period. The trained model can then be deployed for online prediction in Stage III, which uses data fusion to further correct GNN predictions based on residuals between predictions and observations. We describe these steps
in details below.





## 3.1 Graph-based surrogate modeling for river networks

The GNN surrogate modeling framework used in this work is adapted from GraphWaveNet (GWN) (Wu et al., 2019), which consists of two types of interleaved layers, namely, GNN layers for spatial learning and temporal convolution network (TCN) layers for temporal learning. A schematic plot of the adapted GWN design is provided in Fig. 2.

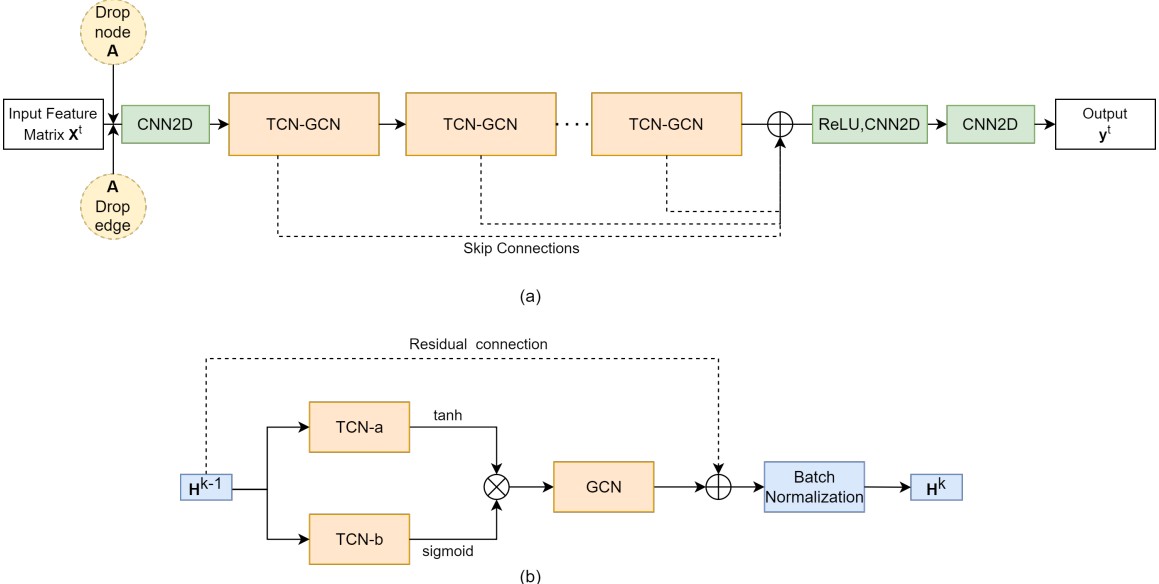

**Figure 2.** (a) GraphWavenet used in this work consists of a number of temporal convolution network and graph convolution network (TCN-GCN) modules for progressively encoding input data $\mathbf{X}^t$; the outputs of each TCN-GCN are skip-connected to improve learning; $1 \times 1$ kernel convolutional neural network (CNN) layers are used as linear transformation layers; optionally, nodes and edges in the adjacency matrix $\mathbf{A}$ are randomly masked for improving training and for graph coarsening (i.e., DropNode and DropEdge); (b) the internal design of each TCN-GCN module, which consists of interleaved TCN and GCN layers to process information from one hidden layer to the next. The layer input and output are connected via residual connections to improve training.

In general, the graph-based learning seeks to learn a low-dimensional representation of input data through recurrent information aggregation and propagation steps. At the node level, a basic GNN layer may be written as (Bronstein et al., 2021)

$$\mathbf{h}_v^{(k)} = f\left(\mathbf{h}_v^{(k-1)}, \bigoplus_{u \in \mathcal{N}_v} \left(\psi(\mathbf{h}_u, \mathbf{h}_v)\right)\right), \tag{1}$$

where $\mathbf{h}_v^{(k)} \in \mathbb{R}^{D_k}$ denotes the embedding or hidden state of node $v$ at the $k$-th layer, $D_k$ is the output dimension

of the $k$-th hidden layer and $\mathbf{h}_v^{(0)} = \mathbf{x}_v$; $\bigoplus$ is an aggregation operator; $\psi$ is a learnable function parameterizing the messaging passing between node $v$ and its neighbors $u \in \mathcal{N}_v$; and $f(\cdot)$ is an activation function (e.g., the Rectified





Linear Unit function or ReLU). Most GNNs differ by how $\bigoplus$ and $\psi$ are chosen. For example, in the GCN layer that is used within GWN, the following aggregation scheme is used, (Kipf and Welling, 2016)

$$\mathbf{h}_v^{(k)} = f\left(\mathbf{W}_1^{(k)}\mathbf{h}_v^{(k-1)} + \mathbf{W}_2^{(k)}\sum_{u\in\mathcal{N}_v}\mathbf{h}_u^{(k-1)} + \mathbf{b}^{(k)}\right), \tag{2}$$

where the weighted sum of the hidden states of neighboring nodes (2nd term in parentheses) is added to the node's embedding from the previous layer (1st term in parentheses). Here $\mathbf{W}_1^{(k)}$ and $\mathbf{W}_2^{(k)}$, both $\in \mathbb{R}^{D_k \times D_{k-1}}$, are learnable weight matrices that determine the influences of node features and node neighbors, respectively; and $\mathbf{b}$ is a learnable bias term added to improve training. At the graph level, the graph convolution operation in Eq. 2 may be written in a simplified matrix form as (Kipf and Welling, 2016)

$$\mathbf{H}^{(k)} = f\left(\tilde{\mathbf{A}}\mathbf{H}^{(k-1)}\mathbf{W}^{(k)}\right), \tag{3}$$

where $\tilde{\mathbf{A}} = \mathbf{A} + \mathbf{I}$ is the adjacency matrix with self-loop (i.e., self-pointing links are added), $\mathbf{I}$ is the identity matrix, $\mathbf{H}^{(k)} \in \mathbb{R}^{N \times D_k}$ includes all hidden-state node vectors in the $k$-th layer, and $\mathbf{W}^{(k)} \in \mathbb{R}^{D_{k-1} \times D_k}$ is a learnable weight matrix. In practice a normalized form of adjacency matrix is often used in lieu of Eq. 3 to improve numerical stability (Kipf and Welling, 2016),

$$\hat{\mathbf{A}} = \tilde{\mathbf{D}}^{(-1/2)}\tilde{\mathbf{A}}\tilde{\mathbf{D}}^{(-1/2)}, \tag{4}$$

where $\tilde{\mathbf{D}}$ is a diagonal matrix containing the node degrees of $\tilde{\mathbf{A}}$.

For temporal learning, GWN adopts TCN, which uses a dilation factor to exponentially increase the receptive field of a CNN filter, thus allowing the capture of long-range dependencies using CNN (Wu et al., 2019; Zhang et al., 2020). Two TCNs (TCN-a and TCN-b in Fig. 2) are used in parallel to form a gated TCN block, as proposed by
Dauphin et al. (2017),

$$\mathcal{X}^{(k)} = g(\Theta_1^{(k-1)}\mathcal{X}^{(k-1)} + \mathbf{b}_1^{(k-1)}) \odot \sigma(\Theta_2^{(k-1)}\mathcal{X}^{(k-1)} + \mathbf{b}_2^{(k-1)}), \tag{5}$$

where the $g(\cdot)$ function updates the hidden state using outputs from the previous layer $\mathcal{X}^{(k-1)} \in \mathbb{R}^{N \times D_{k-1} \times T_B}$, $\sigma(\cdot)$ is a gate function that regulates information flow from one layer to the next, $\Theta_i$ and $\mathbf{b}_i$ are learnable weight matrices and bias terms, and $\odot$ is the element-wise multiplication operator. In GWN, `tanh` is used for $g(\cdot)$ and
`sigmoid` is chosen for $\sigma(\cdot)$. Multiple TCN-GCN modules are then stacked to learn spatial and temporal embeddings progressively. To improve ML learning, the input and output of each TCN-BCN module are connected for residual learning, and the outputs of all TCN-GCN modules are skip connected to the final output layer (see Fig. 2). Finally, $1 \times 1$ kernel CNN layers are used to condense the tensor dimensions and generate the desired outputs.

A well-known issue with the standard GNNs is oversmoothing, which happens when node-specific information is
smoothed out after several rounds of message passing (Li et al., 2018). This can be especially problematic when





features from a node's neighboring nodes become dominant, overshadowing the features of the node itself (Hamilton, 2020). Oversmoothing is a main reason behind the shallow design of many GNN. In the literature, two heuristic strategies have been proposed to mitigate oversmoothing, namely, node dropping and edge dropping. The former strategy randomly masks out a number of nodes in the adjacency matrix during each iteration of training, while

the latter strategy randomly drops a fraction of node edges. The DropEdge scheme, originally proposed by Rong et al. (2019), produces varying perturbations of the graph connections and, thus, can be seen as a data augmentation technique for training GNNs. In contrast, DropNode may be seen as both a training technique and a data imputation strategy—a GNN trained only on a subset of nodes can be used to predict nodes not seen during training. Previously, node masking was used for solving the Prediction at Ungauged Basins (PUB) problem (Sun et al., 2021). In this

work, we further demonstrated its use for solving the Prediction at Unmodeled Nodes (PUN) problem at the basin scale. We implemented both of these features as options within GWN (see Fig. 2).

As mentioned before, the GWN model is first trained using NWM outputs (pretraining) and then the network weights are fine-tuned using observation data. The fine-tuning step is often adopted in physics-guided ML studies to reduce biases of process-based models (Ham et al., 2019; Andersson et al., 2021).

## 3.2   Data fusion

After fine-tuning, the surrogate model may still be subject to errors resulting from ML approximations and from the process-based NWM. New observation data, when available, can be used to further reduce the surrogate model prediction error through a post-processing step. Our goal is closely related to that of the hydrologic post-processing, which seeks to establish a statistical relationship between model outputs and observations (Li et al., 2017). In

computer science, such post-processing is also related to label propagation, referring to assigning class labels to unclassified data using known data labels (Zhu and Ghahramani, 2002).

Let $\mathbf{y}$ and $\hat{\mathbf{y}}$ denote the true and predicted values. In reality, $\mathbf{y}$ can only be accessed at a limited number of observation nodes. Thus, $\mathbf{y}$ is partitioned into two parts corresponding to observed (labeled) and unobserved (unlabeled) values, $\mathbf{y} \doteq [\mathbf{y}_L^T, \mathbf{y}_U^T]^T$. Further, assume $\mathbf{y}$ is multi-Gaussian, $\mathbf{y} \sim \mathcal{N}(\boldsymbol{\mu}, \boldsymbol{\Sigma})$, with mean $\boldsymbol{\mu}$ and covariance matrix $\boldsymbol{\Sigma}$.

Streamflow values, which typically follow non-Gaussian distributions, can be projected into the normal space via a normal transform technique (Li et al., 2017). As a matter of fact, this normal transform generally improves ML training and, thus, should be done as part of the data preprocessing before ML training starts.

The joint distribution of $\mathbf{y}$ in terms of its subsets $\mathbf{y}_L$ and $\mathbf{y}_U$ may be written as (Bishop, 2006),

$$p(\mathbf{y}_L, \mathbf{y}_U) = \mathcal{N}\left( \begin{pmatrix} \boldsymbol{\mu}_L \\ \boldsymbol{\mu}_U \end{pmatrix}, \begin{pmatrix} \boldsymbol{\Sigma}_{LL} & \boldsymbol{\Sigma}_{LU} \\ \boldsymbol{\Sigma}_{UL}, & \boldsymbol{\Sigma}_{UU} \end{pmatrix} \right). \tag{6}$$





It can be shown the conditional probability distribution of $\mathbf{y}_U$ given $\mathbf{y}_L$, namely, $p(\mathbf{y}_U \mid \mathbf{y}_L)$, is also multi-Gaussian, for which the conditional mean $\boldsymbol{\mu}_{U|L}$ and covariance $\boldsymbol{\Sigma}_{U|L}$ are (Bishop, 2006)

$$
\begin{aligned}
\boldsymbol{\mu}_{U|L} &= \boldsymbol{\mu}_U + \boldsymbol{\Sigma}_{UL}\boldsymbol{\Sigma}_{LL}^{-1}(\mathbf{y}_L - \boldsymbol{\mu}_L) = \boldsymbol{\mu}_U - \boldsymbol{\Gamma}_{UU}^{-1}\boldsymbol{\Gamma}_{UL}(\mathbf{y}_L - \boldsymbol{\mu}_L), \\
\boldsymbol{\Sigma}_{U|L} &= \boldsymbol{\Sigma}_{UU} - \boldsymbol{\Sigma}_{UL}\boldsymbol{\Sigma}_{LL}^{-1}\boldsymbol{\Sigma}_{LU} = \boldsymbol{\Gamma}_{UU}^{-1},
\end{aligned}
\tag{7}
$$

where $\boldsymbol{\Gamma}$, known as the precision matrix, is the inverse of the covariance matrix. In the regression problem considered here, $\boldsymbol{\mu}_U$ may represent the ML estimates at unobserved locations, $\hat{\mathbf{y}}_U$, and $\mathbf{y}_L$ may represent gage observations,

then Eq. 7 forms the basis of updating the ML predictions $\hat{\mathbf{y}}_U$ through observations $\mathbf{y}_L$. To adapt the residual propagation for GNNs, Jia and Benson (2020) proposed the following parameterization of the precision matrix,

$$
\boldsymbol{\Gamma} = \beta(\mathbf{I} - \alpha\mathbf{S}),
\tag{8}
$$

where $\mathbf{S} = \mathbf{D}^{-1/2}\mathbf{A}\mathbf{D}^{1/2}$ is similar to the normalized adjacency matrix defined in Eq. 4 but with self-loops removed; $\beta$ and $\alpha$ are learnable shape parameters, the former controls the residual magnitude and the latter reflects the cor-

relation structure. Later, the same authors proposed a residual propagation form involving a single hyperparameter $\omega$ (Jia and Benson, 2021),

$$
\hat{\mathbf{y}}_{U|L} = \hat{\mathbf{y}}_U - (\mathbf{I} + \omega\mathbf{N})_{UU}^{-1}(\mathbf{I} + \omega\mathbf{N})_{UL}\mathbf{r}, \ \mathbf{r} \doteq (\mathbf{y}_L - \hat{\mathbf{y}}_L) \ \text{and} \ \mathbf{N} \doteq \mathbf{I} - \mathbf{S}
\tag{9}
$$

where the labeled and unlabeled parts of $\mathbf{I} + \omega\mathbf{N}$ are extracted by using mask matrices, $\mathbf{r}$ is defined as the residual vector between observations and ML predictions. Eq. 9 is the form of data fusion used in this study, which is model

agnostic. The hyperparameter $\omega$ may be obtained by cross validation. The only time-varying part in Eq. 9 is $\mathbf{r}$ and other matrix terms can be calculated offline, thus the update can be applied efficiently in real time.

We remark that the conditional mean and covariance shown in Eq. 7 are generally related to the Gaussian Process regression (Rasmussen and Williams, 2006) and has been used in the hydrologic post-processing literature, for example, in the General Linear Model Post-Processor in (Ye et al., 2014). However, the main difference is the data

fusion is extended to operate on graphs via Eq. 9. An advantage of the residual propagation approach taken here is that it allows consideration of spatial correlation among node prediction errors while respecting the graph topology. Intuitively, we expect that unobserved nodes adjacent to a gauged node should share similar spatial and temporal patterns. The data fusion approach adopted here is different from the data assimilation method in (Jia et al., 2021), in which a prediction is made by using the features of a node and its neighboring nodes, but not directly considering

predictions of the neighboring nodes, thus limiting the use of spatial information (Jia and Benson, 2020).

## 4 Study Area and Experiment Design

### 4.1 Description of the study area

The algorithms and workflow described in Section 3 are generic. For demonstration purposes, we consider the East-Taylor Watershed (ETW), a HUC-8 watershed (drainage area 1984.7 km$^2$) that lies within the Gunnison River



basin (HUC-4) in the southern Rocky Mountains of Colorado, U.S. (Fig. 3). The ETW is representative of the
high-altitude river basins in the upper Colorado River basin. Elevation of the watershed ranges approximately from
2,440 to 4,335 m (McKay et al., 2012). Climate of the watershed is defined as continental, subarctic climate with
long, cold winters and short, cool summers (Hubbard et al., 2018). Annual precipitation ranges from 1350 mm/yr
in the high-elevation headwaters region of East River to about 400 mm/yr near the basin's outlet, and the majority
of precipitation falls as snow (see Fig. S1a in Supporting Information (SI)); annual temperature is in the range of -3
to 1 °C in the area (Fig. S1b) (PRISM Climate Group, 2022).

The ETW encompasses two alpine rivers, the East River in the west and the Taylor River in the east, both flowing
into the Gunnison River in the south which, in turn, serves as a main tributary of the Colorado River, contributing
about 40 percent of its streamflow (Battaglin et al., 2011). The East River watershed is mostly undeveloped, other
than the city of Crested Butte (population 1339) and the ski resort area that are located near the middle course of
the river (Bryant et al., 2020). The Taylor River is dammed by Taylor Park Dam (storage capacity about 0.13 km$^3$)
in the middle (Bureau of Reclamation, 2022).

Snowmelt provides the main source of runoff in the watershed, with peak discharge occurring between May
and July; baseflow conditions prevail from August until the winter freeze (Bryant et al., 2020). Like many other
snow-dominated systems in the western U.S., streamflow pattern in the ETW is influenced by frequent droughts
and heatwaves in recent years (Winnick et al., 2017), and is likely to undergo further changes with the projected
lower snowfall and earlier snowmelt under future climate conditions (Davenport et al., 2020). Globally, tremendous
interests exist in the hydroclimate modeling community to understand and predict streamflows in snow-dominated
regions under global environment change (Barnett et al., 2005; Qin et al., 2020).

Five USGS gages in the watershed are identified to have continuous records (open circles in Fig. 3) and are used
as the source for fine-tuning and data fusion in this work. Notably Gage 09107000 and Gage 09109000 are located
immediately upstream and downstream of the Taylor Park Dam. The mean annual flow at the outlet of East River
(Gage 09112500) is 9.3 m$^3$/s, and at the outlet of Taylor River (Gage 09110000) it is 9.2 m$^3$/s, both are estimated
based on 110 years of data from 1911–2021 (USGS, 2022a). An extra USGS gage (red circle) with incomplete record
is also identified for validation purposes.

The ETW contains 23 subbasins (dark lines in Fig. 3) at the HUC-12 level and a total of 552 NHDPlus flowlines or
reaches (dark blue polylines in Fig. 3). The NHDPlus reach lengths range from 0.047 to 11.48 km, the stream order
ranges from one (the headwater tributaries) to five (Taylor River downstream of the reservoir), and the drainage
area of reaches ranges from 0.027–24.99 km$^2$. Thus, ETW presents an interesting study area from the perspective
of river network modeling: it encompasses two contrasting flow regimes, the East River that is under natural flow
conditions and the Taylor River that is subject to human intervention; it also includes a meaningful degree of spatial
heterogeneity that is representative of a snow-dominated, mountainous watershed.



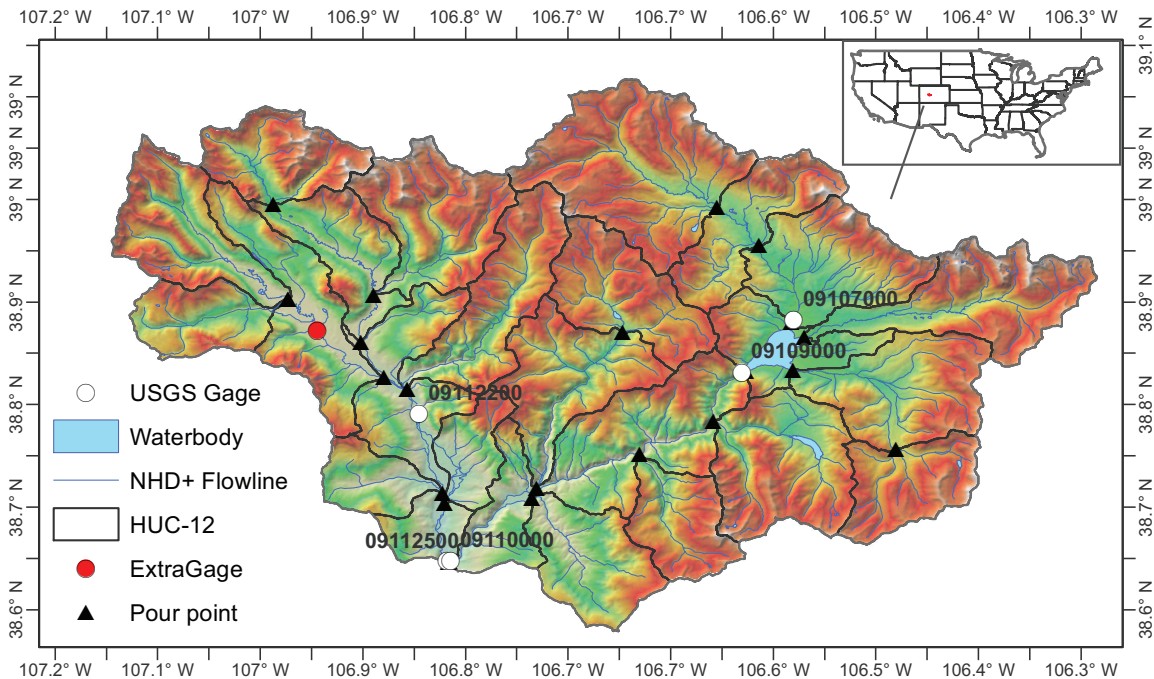

**Figure 3.** Shaded relief map of the East-Taylor watershed (HUC-8 code 14020001), overlaid by the NHDPlus flowline (solid blue lines), five USGS gage locations (open circles) used for fine-tuning and data fusion, the HUC-12 subbasin boundaries (solid dark lines) and pour points (triangles), and the surface waterbody (light blue) layer. An extra gage (red circle) is held for additional validation

### 4.2 Experimental design and model training

We performed a series of ML experiments to address the two objectives of this study, namely, evaluating the role of physical realism in river network representation and the efficacy of graph-based data fusion. The study period is set to 1993/01–2018/12, the same as the NWM v2.0's retrospective simulation period. All scripts are written in Python and, in particular, PyTorch (Paszke et al., 2019) is used to develop all GNN and data fusion codes. The open-source GWN code (Wu et al., 2019) and GNN data fusion code (Jia and Benson, 2020) are adapted for the problem at hand. The train/validation/testing split used is (0.7, 0.15, 0.15). Streamflow data is projected to normal

space by using a power transformer scaler proposed in (Yeo and Johnson, 2000) and available from $\mathtt{scikit-learn}$ (Pedregosa et al., 2011), in which the scaler is first fit to the training data and then applied to the remainder of the dataset. An inverse transform is then applied at the end to convert the results back to the original space.

The node-based ETW river network is constructed based on the NHDPlus flowlines and associated reach IDs (COMIDs) by following the procedures described under Section 2.3. A cutoff threshold is applied to extract a subset

of river reaches. Thresholding is a common practice in climate networks to help reveal dominant spatiotemporal structures (Donges et al., 2009a, b; Malik et al., 2012). Here all reaches with a medium flow value less than 0.01





m$^3$/s (approximately at the 30th percentile of median flow distribution of all nodes) are trimmed from the network, keeping 295 out of a total of 552 nodes. In the resulting adjacency matrix, two nodes are connected if an NHDPlus flowline of "streamriver" type exists between them. Reaches falling on the water bodies (light blue polygons in Fig. 3)

are not included in the river network. We used undirected and unweighted adjacency matrices. As an ablation study, we also considered NWM v2.1 data, which resulted in a 348-node network after trimming. The different numbers of node probably reflect differences in model parameterization and forcing data between the two NWM versions (see Section 2). The HUC-12 subbasin-based ETW river network is constructed by using the information in the NHDPlus pour point layer (Price, 2022) for connecting each subbasin to its neighbors.

Unless otherwise noted, we train the model with a lookback period of 30 days, which is sufficient for the current case when simulated antecedent flows are also used as predictors to drive streamflow predictions. The forecast horizon is one-day ahead. The number of TCN-GCN blocks used is 7, the kernel used in TCN layers is $1 \times 4$, the number of filters used in dilation and residual blocks are both 32. We use AdamW, a modified version of the Adam gradient-based optimizer, for training the network (Loshchilov and Hutter, 2018). The mini-batch size is 30.

When using the DropEdge option, we randomly enable 80 percent of the edges in the adjacency matrix at the beginning of each epoch. This is done by flattening the adjacency matrix into a vector, performing random permutation, and taking the first 80 percent of connection. When the DropNode option is on, we use a mask matrix to mask the indices of nodes to be dropped. The training parameter selection is largely based on our previous experience (Sun et al., 2021). For each GWN configuration, 10 different models are trained by initializing with different random

seeds. For pretraining, the loss function used is the mean absolute error (MAE) or $L_1$ norm between ML predictions and NWM outputs, while for fine-tuning, the loss function is the $L_1$ norm between ML predictions and observed streamflow at the five gage locations. Pretraining is done for 60 epochs with a learning rate of 0.0005. Starting with the weights of a pretrained model, fine-tuning is done for another 15 epochs but with a smaller learning rate of $1e - 5$. Training time is around 1.3 min wallclock time per epoch for the 295-node models, and 14 sec per epoch for

the 23-node models, on the same Nvidia V100 GPU.

We quantify the performance of trained models on test data using three metrics, namely, normalized root mean square error (NRMSE), Kling–Gupta efficiency (KGE) (Gupta et al., 2009), and Pearson's correlation (R). Definitions of the metrics and $L_1$ norm are given in the Appendix. The hyperparameter $\omega$ in Eq. 9 is selected through leave-one-out cross validation (LOOCV). Specifically, the $\omega$ value is varied in the range (50, 5000) with a step size of 50.

For each $\omega$, we perform LOOCV by using four of the five gages for residual propagation, and calculating the MAE on the holdout gage. The $\omega$ that gives the minimum mean MAE across all gages is selected for data fusion.



## 5 Results

### 5.1 Pretraining

We first demonstrate the efficacy of GWN and its variants for approximating the NWM (i.e., the pretraining stage).
Three sets of GWN models are trained. The first set uses the original GWN configuration with the self-loop adjacency
matrix $\tilde{\mathbf{A}}$ (defined in Eq. 3) corresponding to the actual ETW network (GWN-O), the second set uses the same
$\tilde{\mathbf{A}}$ as input but is trained by activating the DropEdge feature to randomly remove node links (GWN-DropEdge),
and the third set is trained using only a subset of nodes in $\tilde{\mathbf{A}}$ (GWN-Impute). Thus, physical realism is gradually
reduced across the three sets of experiments. The $\tilde{\mathbf{A}}$ used in the first two sets corresponds to the trimmed ETW
network containing 295 nodes, while the third set uses the HUC-12 pour point set as nodes (see Fig. 3).

Table 1 summarizes the performance metrics of all three models in KGE, NRMSE, and R. Results suggest the
GWN-O achieves high scores under all three metrics. GWN-DropEdge gives almost the same performance. GWN-
Impute, which is trained using less than one-tenth of the nodes used in the other two models, also achieves a
reasonable performance. Fig. 4 shows the empirical cumulative distribution function (ECDF) and node-level maps
of KGE and NRMSE that are obtained by the GWN-O surrogate model. The KGE is close to 1.0 (NRMSE close to
0.0) on the mainstems of East and Taylor rivers, but drops (increases) slightly in several tributaries of the Taylor
River in the middle and southeast part of the watershed. Performance at subbasins disconnected by the Taylor
Park Dam does not appear to be strongly affected. All reaches adjacent to the reservoir show high KGE. This is
interesting, suggesting the antecedent NWM outputs provide sufficient node-level information for the GNN to learn.
As an ablation study, we trained a GWN-O model using only Daymet forcing data while keeping all other config-
urations unchanged. Results, shown in Fig. S2 of the SI, suggest the performance of the data-driven surrogate model
is deteriorated, especially in the mid-range of both rivers. One possible explanation is without using the antecedent
NWM outputs as predictors, the data-driven models may either require a much longer lookback period to achieve
meaningful results (Kratzert et al., 2019a), or simply cannot explain all variations in the NWM outputs. The latter
reason points to the inherent difficulty in learning the data-driven, input-output mapping in this snow-dominated
watershed. Previously, Ma et al. (2017) evaluated the performance of NOAH-MP, which is the land model in WRF-
Hydro, in modeling the snow cover function (SCF). By definition, SCF is the fraction of a grid cell covered by snow,
and provides an indirect measure of snow mass and snow depth. They found that the modeled SCF agrees well with
the gridded SCF product derived from Moderate Resolution Imaging Spectroradiometer (MODIS), with relative
biases varying from 4% in the snow accumulation phase to 14% in the melting phase. The authors attributed the
good performance to the SCF scheme, the use of a vegetation canopy snow interception module, and the multilayer
snowpack representation implemented in NOAH-MP (Ma et al., 2017). Thus, these sophisticated parameter schemes
in NWM may not be explained by using Daymet forcing alone.

We compare the KGE between GWN-O and the other two models in Fig. 5. The performances of GWN-O and
GWN-DropEdge are similar, only slight KGE differences are noted in the headwater tributaries of Taylor River,





indicating the GWN models are relatively robust to perturbations induced through dropping edges. This is again because the same NWM data is behind GWN-O and GWN-DropEdge and strong auto-correlation exists at the node level. Between GWN-O and GWN-Impute, GWN-O outperforms in the headwater subbasins. Nevertheless, the accuracy of GWN-Impute is high on the mainstems.

**Table 1.** Performance metrics of pretrained GraphWaveNet (GWN) surrogate models. GWN-O is the original model of Wu et al. (2019), GWN-DropEdge implements random edge sampling, GWN-Impute is trained on a subset of 23 nodes corresponding to the HUC-12 pour points. All results reported are based on ensemble averages from 10 models trained with different random seeds.

| Model | Median | Mean | Max | Min |
|---|---|---|---|---|
| KGE, range $(-\infty, 1]$ | | | | |
| GWN-O | 0.978 | 0.947 | 0.998 | 0.574 |
| GWN-DropEdge | 0.978 | 0.947 | 0.998 | 0.584 |
| GWN-Impute | 0.975 | 0.915 | 0.997 | 0.445 |
| NRMSE, range $[0, \infty)$ | | | | |
| GWN-O | 0.188 | 0.256 | 1.126 | 0.021 |
| GWN-DropEdge | 0.186 | 0.262 | 1.132 | 0.023 |
| GWN-Impute | 0.214 | 0.300 | 1.455 | 0.039 |
| R, range $[-1, 1]$ | | | | |
| GWN-O | 0.991 | 0.985 | 1.000 | 0.877 |
| GWN-DropEdge | 0.990 | 0.984 | 1.000 | 0.912 |
| GWN-Impute | 0.990 | 0.982 | 0.999 | 0.906 |

To elucidate why GWN-Impute gives a good performance on ETW, we plot the node correlation heatmap in Fig. 6. Subbasins 1–13 are on the Taylor River side, while subbasins 14–23 are on the East River side of the ETW (Fig. 6a). An immediate observation is the block structure along the diagonal of heatmap in Fig. 6c, suggesting inside each subbasin strong cross-node correlations exist. Specifically, subbasin 13, the most downstream basin on the Taylor River side, exhibits the highest inner-basin correlation, while the upstream headwaters basins (subbasins 1

and 2) show more inner-basin variations. Subbasin 13 also shows relatively strong cross-basin correlations with other subbasins on both the East River and Taylor River sides. This is because subbasin 13 is at the confluence of the two rivers, thus reflecting information passed from the upstream of both rivers. Similarly, the downstream basins on the East River side, namely, subbasins 22 and 23, also show relatively strong inter-basin correlations with other basins on the East River side. In contrast, isolated remote subbasins (e.g., #14 on East River side, and #1 and #11 on Taylor

River side as labeled below Fig. 6c do not exhibit strong inter-basin correlations. These observations are further corroborated using the graph betweenness centrality, defined as the number of shortest paths that pass through a



**Figure 4.** Test performance metrics obtained using the pretrained 295-node GraphWaveNet (GWN-O) model. (a)&(b) empirical cumulative distribution function (ECDF) and node map of KGE; (c)&(d) ECDF and node map of NRMSE. Results are obtained based on the ensemble mean of a 10-member ensemble. Gage locations are shown as open circles on node maps.
.



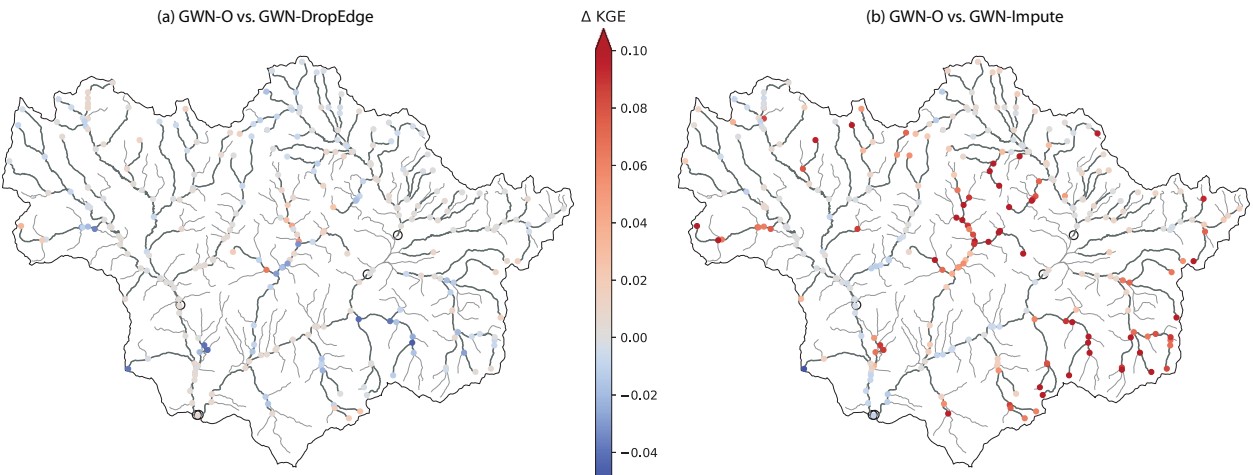

**Figure 5.** Node-level KGE comparison between the pretrained GWN-O and that of (a) GWN-DropEdge and (b) GWN-Impute.

node. Thus, nodes with a high betweenness centrality tend to have more influence on information propagation in the network (Donges et al., 2009b). Fig. 6b shows reaches in the downstreams of East and Taylor rivers have higher betweenness centrality values than the upstream nodes. The heatmap analysis reveals the model parameter structure and the degree of freedom of the underlying NWM which, in turn, determine how well the graph coarsening process may work. Essentially, in this case the pour-point based ML imputation is a physics-informed interpolation process. We expect the information content is richest at the pour point of each subbasin, thus a surrogate model trained using only pour-point information may sufficiently capture the dominant flow dynamics in the watershed and satisfactorily interpolate to all unmodeled points.

We conducted an ablation study using the 348-node network corresponding to NWM 2.1 (see also Section 4.2). Results are shown in Fig. S3. The newly added nodes are low-flow nodes appearing at the headwaters and small tributaries in subbasins. In this case, the median of the KGE is 0.974 and the mean is 0.937. Comparing to the model trained using NWM v2.0, the performance in subbasins #8, #10, #11 are significantly improved, however, the performance in headwater subbasins #1 and #2 on the Taylor River side also show some deterioration. This suggests the effect of model calibration between NWM versions may not be uniform across the model domain.

Experiments presented thus far provide useful insights on how much physical realism is needed when implementing GNNs for the purpose of river network surrogate modeling. It depends on the watershed hydroclimatic and physiographic attributes, the parameter structure of the underlying process-based model, and the objective of study. In the current case, we show that GWN models built on a coarsened graph give comparable performances as those built on more fine-grained representations of the river network, requiring only a fraction of training time. If the main objective of surrogate modeling is to capture the simulated streamflow patterns in the mainstem of a river (e.g., for



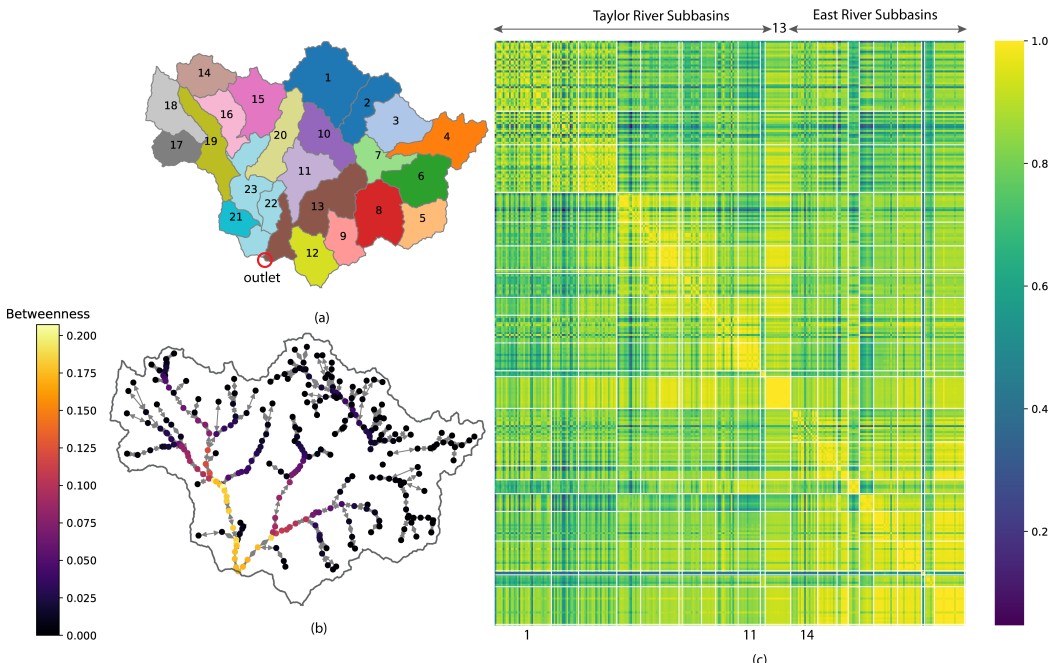

**Figure 6.** Node correlation heatmap generally exhibits a block structure within subbasins: (a) subbasin boundary map, where subbasins 1–13 are on the Taylor River side and subbasins 14–23 are on the East River side; (b) node correlation heatmap, where subbasin nodes are separately by white solid lines

estimating flood peaks), then all variants of the GWN models presented here should suffice. On the other hand, if the main objective is to simulate snowmelt and runoff in low-flow headwater basins, then more physical realism is required in the river network. These findings also shed light on scaling up the GNNs for modeling large river basins.

## 5.2 Fine-tuning

After pretraining, we fine-tune the GWN models by following the procedure described in Section 4.2. Table 2 reports the KGE and R of NWM, pretrained and fine-tuned GWN-O models against the five USGS gages (see Fig. 3 for their locations) over the testing period. The corresponding hydrographs are presented in Fig. 7. The KGE values of NWM simulations are relatively low, only Gage 09107000 (upstream of Taylor Park Dam) and Gage 09110000 (at pour point Taylor River) have KGE values greater than 0.5. The pretrained GWN-O model reports similar KGE values as the NWM, which is by design. The fine-tuned GWN-O model, which is trained using the historical observations falling in the training period, improves the KGE values slightly over most gages, except for Gage 09112500 located in the mid-stream of East River. We also calculate the KGE values for winter season (NDJF months) and summer season (MJJA months) separately. The fine-tuned model captures the low flow on East River relative well (Gage





**Table 2.** Comparison of the KGE and R of NWM, pretrained GWN-O, fine-tuned GWO-O against the USGS gage data. The last four columns show the values for winter and summer seasons.

| USGS Gage | KGE | | | | | | |
|---|---|---|---|---|---|---|---|
| | NWM | GWN-O Pretrain | GWN-O Fine-tune | NWM (winter) | GWN-O Fine-tune (winter) | NWM (summer) | GWN-O Fine-tune (summer) |
| 09107000 | 0.563 | 0.559 | 0.596 | 0.169 | 0.205 | 0.467 | 0.485 |
| 09112200 | 0.332 | 0.328 | 0.348 | 0.192 | 0.625 | 0.266 | 0.258 |
| 09112500 | 0.355 | 0.348 | 0.341 | -0.076 | 0.741 | 0.289 | 0.248 |
| 09109000 | 0.425 | 0.425 | 0.624 | -1.418 | -0.146 | 0.323 | 0.524 |
| 09110000 | 0.616 | 0.610 | 0.665 | -0.632 | 0.148 | 0.534 | 0.614 |
| | R | | | | | | |
| 09107000 | 0.877 | 0.876 | 0.851 | 0.698 | 0.318 | 0.855 | 0.771 |
| 09112200 | 0.904 | 0.900 | 0.961 | 0.348 | 0.660 | 0.898 | 0.953 |
| 09112500 | 0.901 | 0.899 | 0.957 | 0.074 | 0.756 | 0.894 | 0.948 |
| 09109000 | 0.554 | 0.554 | 0.782 | -0.471 | -0.112 | 0.442 | 0.650 |
| 09110000 | 0.706 | 0.715 | 0.868 | -0.337 | 0.195 | 0.618 | 0.788 |

09112200 and 09112500). However, all peak flows are underestimated, as can be seen from Fig. 7. In comparison, the fine-tuned model generally makes more improvement on R, except at Gage 09107000. The summer correlation values are higher than the winter values.

Overall, although fine-tuning of the GWN-O only leads to mild performance improvements in this case, the bias corrections, especially in the phase of the time series, are important for the subsequent data fusion step. In this

work, we mainly utilized streamflow data, but other types of Earth observation data may also be integrated in the fine-tuning step to further improve model performance.

### 5.3 Data fusion

We investigate the efficacy of data fusion on the five USGS gages. The value of hyperparameter $\omega$ is determined according to the LOOCV procedure described under Section 4.2. For the fine-tuned 295-node GWN-O model, the

optimal $\omega$ value is found to be 1500. Fig. 8 compares NWM v2.0, corrected GWN-O, and observed streamflow time series for the testing period, in which the KGE and correlation (R) between the corrected GWN-O and observations are shown in the subplot titles for each gage.

After data fusion, R becomes greater than 0.85 at all gage locations. In terms of KGE, the LOOCV data fusion has the greatest impact on Gage 09109000, achieving a value of 0.944. It also improves the two East River gages



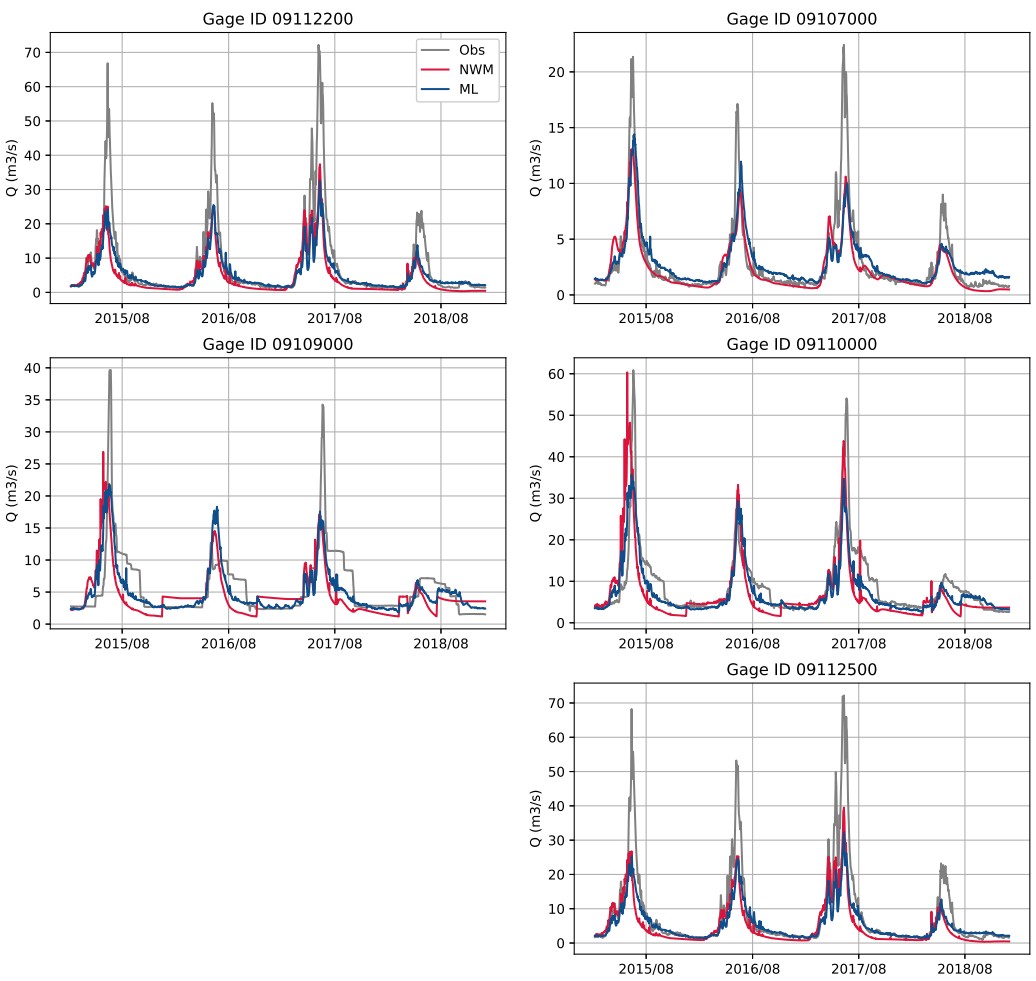

**Figure 7.** Hydrographs simulated by NWM (dark blue) and GWN-O (ML, dark red) vs. the USGS data (Obs, gray) over the testing period.





significantly (i.e., Gage 09112500 and 09112200). It has little effect on 09107000, which is upstream of the Taylor Park Reservoir. It has a slightly negative impact on Gage 09110000, which is at the outlet of Taylor River. In this case, Gage 091125000 and 09110000 are very close to each other (indirectly connected via the confluence node between East and Taylor rivers), but reflecting different flow patterns. Specifically, Gage 09110000 is affected by the reservoir releases, as can be seen from the zigzag pattern in Fig. 8, while Gage 091125000 is not subject to

such human intervention impact. The current data fusion scheme does not differentiate these different dynamics. However, when all gages are used simultaneously in data fusion, we expect such interference to be reduced.

Fig. 9 shows the data fusion residual map, defined as the flow difference between data fusion and the NWM (i.e., $\mathbf{r}$ in Eq. 9). The data fusion effect is greatest at the most downstream locations of both East and Taylor rivers and then gradually fades out toward the upstream headwaters basins, reflecting the reduction in flow magnitudes when

traversing upstream, as well as the diminished influence of downstream gages.

The effect of data fusion effect is further validated using an extra USGS gage on East River, Gage 385106106571000, that is not part of the model training and residual propagation (see Fig. 3 for its location). This is the only extra gage in ETW that has a meaningful length of records (4475 days) for the study period. Results (Fig. 9) show that data fusion significantly improved streamflow, increasing the KGE from 0.056 to 0.892.

As an ablation study, we applied the same data fusion procedure to another two models, the GWN-Impute model and the GWN-O model trained without using NWM as predictors. The LOOCV results, shown in SI tables S1 and S2, suggest that data fusion in general improves the results at most gage locations. The GWN-Impute model shows comparable performance as the GWN-O model, although it is challenging for the GWN-O trained without using NWM to yield good results. Thus, these results further demonstrate the robustness of a coarsened graph network

for modeling the study area.

## 6    Conclusions

GNNs, comprising nodes and edges and various constructs for graph-based information propagation, provide a conceptually simple and yet powerful ML framework for learning vector hydrography. This work presents a multi-stage, physics-guided ML framework that combines physics-based river network models with GNNs for streamflow

forecasting. In particular, our workflow, consisting of pretraining, fine-tuning, and data fusion stages, leverages existing investment in high-performance river network models (e.g., the U.S. National Water Model) and Earth observation data.

We demonstrated the merits of the GNN-based framework over the East Taylor watershed, a snow-dominated system located in the Upper Colorado river basin. The watershed, which is representative of the snow-dominated

river basins in the western U.S., presents meaningful challenges from both the scientific and ML perspectives. On the science side, significant research interests exist in understanding and predicting snowpack depth and shift in snowmelt timing under projected climate change. On the ML side, challenges remain on scaling graphs for solving

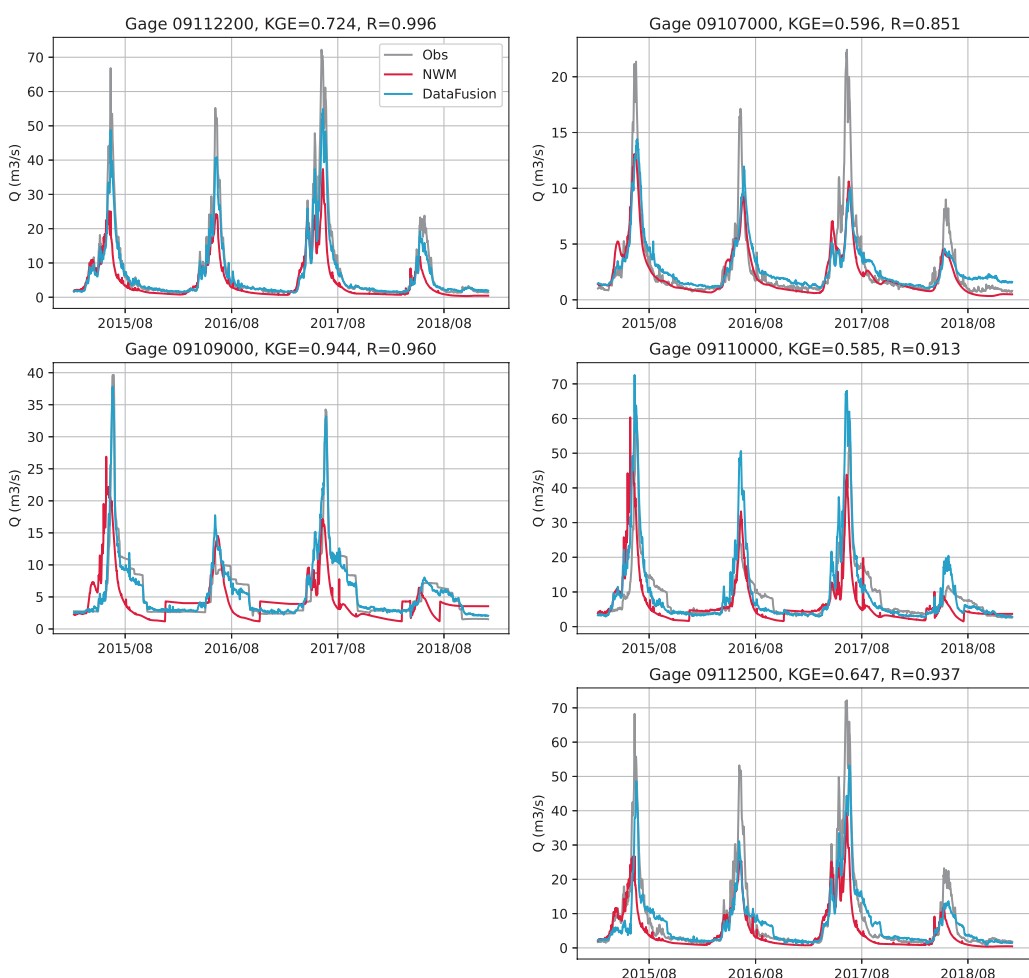

**Figure 8.** Hydrographs simulated by NWM (dark red) and corrected GWN-O (Data Fusion, lightblue) vs. the USGS data (Obs, gray) over the testing period. KGE and R shown in the subplot titles are calculated between data fusion and observations.



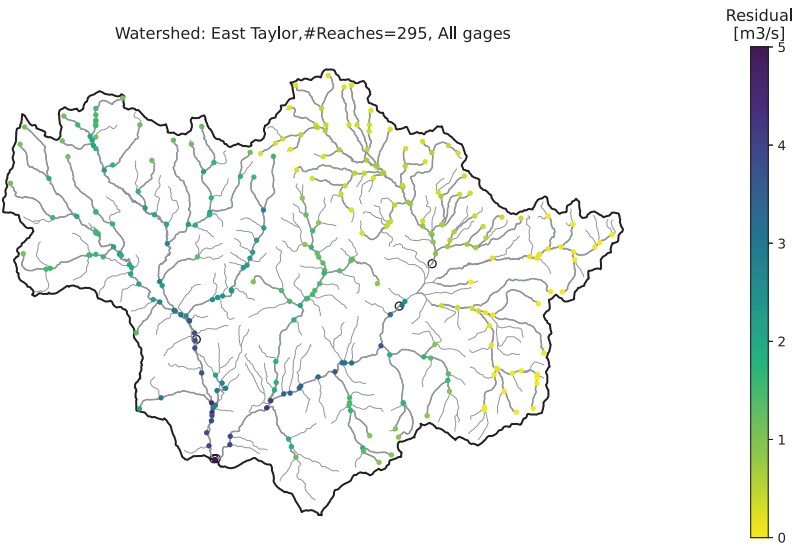

**Figure 9.** Residuals resulting from the data fusion using all five USGS gages. The greatest correction effect is seen at the downstream points.

large-scale graph-based regression problems. We showed that graph coarsening offers a feasible solution by exploiting the parameter structure of the underlying physics-based model. Thus, the designs of GNN and physics-based models 545 can be asymmetric. When the physics-based model is already fine grained, which is the case in our work, the number of nodes in the GNN models can be significantly reduced. This same idea can be further expanded to embrace multiscale and/or multi-fidelity modeling to address different study objectives, such as coupling a fine-mesh network for biogeochemical transport with a coarse-mesh network for streamflow modeling. Heterogeneous graphs may be adopted for those applications (Wang et al., 2019). Finally, we show that graph-based data fusion 550 provides a powerful post-processing tool for "nudging" streamflow observations, allowing error residuals to traverse over the entire network. Thus, this study, although demonstrated over a single watershed, lays the ground for future large-scale river basin analyses.

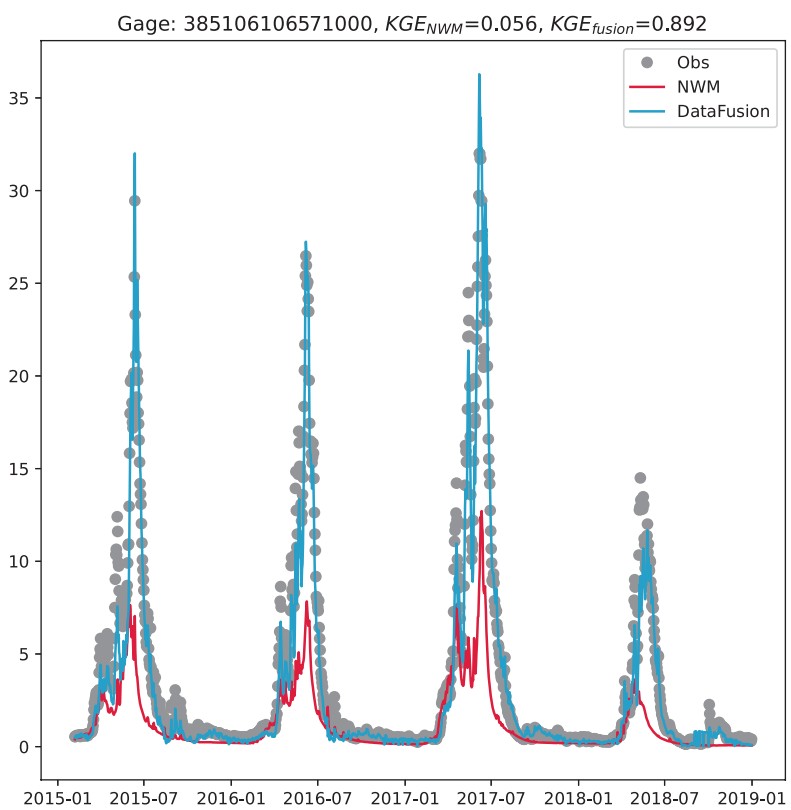

**Figure 10.** The effect of data fusion is validated on Gage 385106106571000 which is not used in any ML training and residual propagation. The KGE is improved from 0.056 to 0.892.

*Code and data availability.* We adapted the open-source GraphWavenet code https://github.com/nnzhan/Graph-WaveNet, data fusion code https://github.com/000Justin000/gnn-residual-correlation, and DropEdge https://github.com/DropEdge/DropEdge in this work. The graph betweenness is generated using NetworkX https://networkx.org.

NWM retrospective simulation (v2.0 and v2.1) data can be downloaded from AWS (https://registry.opendata.aws/nwm-archive). Alternatively, NWM data (v2.0) can be downloaded from Hydroshare ( https://www.hydroshare.org). Daymet data can be downloaded from https://daymet.ornl.gov/. NHDPlus database can be downloaded from EPA's NHDPlus site, https://www.epa.gov/waterdata/get-nhdplus-national-hydrography-dataset-plus-data.



## Appendix A: Definition of performance metrics

The performance metrics used in this work are defined as

$$\text{NRMSE} = \frac{1}{\bar{Q}_{obs}} \sqrt{\frac{\sum_{t=1}^{n} \left(Q_{obs}^t - Q_{sim}^t\right)^2}{n}} \tag{A1}$$

$$\text{R} = \frac{\sum \left(\left(Q_{sim}^t - \bar{Q}_{sim}\right)\left(Q_{obs}^t - \bar{Q}_{obs}\right)\right)}{\sqrt{\sum_{t=1}^{n} \left(Q_{sim}^t - \bar{Q}_{sim}\right)^2 \sum_{t=1}^{n} \left(Q_{obs}^t - \bar{Q}_{obs}\right)^2}} \tag{A2}$$

$$\text{KGE} = 1 - \sqrt{\left(1 - \text{R}\right)^2 + \left(1 - \mu_{sim}/\mu_{obs}\right)^2 + \left(1 - \sigma_{sim}/\sigma_{obs}\right)^2} \tag{A3}$$

$$\text{MAE} = \frac{1}{n} \sum_{t=1}^{n} \mid Q_{obs}^t - Q_{sim}^t \mid \tag{A4}$$

where $Q_{obs}$ and $Q_{sim}$ are observed and predicted values, respectively, $\bar{Q}$ denotes the mean values, and $n$ is the total number of test data. The KGE score combines the linear Pearson correlation (R), the bias ratio $\mu_{sim}/\mu_{obs}$, and the variability ratio $\sigma_{sim}/\sigma_{obs}$ (Gupta et al., 2009). The range of KGE is $(-\infty, 1]$. A KGE value greater than -0.41 indicates the model improves upon the mean flow (Knoben et al., 2019). The range of NRMSE is $[0, \infty)$. The mean absolute error (MAE), or $L_1$ norm, quantifies the absolute difference between simulated and measured values.

*Author contributions.* A.Sun was responsible for the conceptualization, methodology, software, data curation, formal analysis, and writing the original draft. P.Jiang, L.Yang, Y.Xie and X.Chen assisted with the conceptualization, review, and editing of the paper.

*Competing interests.* The contact author has declared that neither they nor their co-authors have any competing interests.

*Acknowledgements.* This work was funded by the ExaSheds project, which was supported by the U.S. Department of Energy (DOE), Office of Science, Office of Biological and Environmental Research, Earth and Environmental Systems Sciences



Division, Data Management Program. Pacific Northwest National Laboratory is operated for the DOE by Battelle Memorial Institute under contract DE-AC05-76RL01830. This paper describes objective technical results and analysis. Any subjective views or opinions that might be expressed in the paper do not necessarily represent the views of the U.S. Department of Energy or the United States Government. A. Sun wants to acknowledge the support of Texas Advanced Computing Center. A. Sun and Z.L. Yang are partly supported by DOE Advanced Scientific Computing Research under Grant DE-SC0022211.




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
