# Peer review of "A graph neural network approach to basin-scale river network learning: The role of physics-based connectivity and data fusion"

_Hydrology and Earth System Sciences, 2022_

## Referee Comment (RC2)

**Review of Manuscript**

**'A graph neural network approach to basin-scale river network learning: The role of physics-based connectivity and data fusion'**

**by Sun et al.**

Dear Editor, dear Authors,

I have reviewed the aforementioned work. My conclusions and comments are as follows:

**1. Scope**

The article is well within the scope of HESS.

**2. Summary**

The authors propose a machine-learning (ML) approach to catchment hydrological modeling in daily resolution, consisting of graph neural networks (GNNs) pretrained on a distributed hydrological model (thus absorbing information about physical processes and subbasin connectivity as encoded in the model), fine-tuned on observed streamflow at available gauges (thus absorbing real-world information not captured by the hydrological model), and finally, if forecasting rather than simulation is the goal, a data-fusion approach (thus absorbing information of observations from the immediate past not captured by GNN modeling based on observed of forcing, but not streamflow). The authors demonstrate the approach at the example of the high alpine, snow-influenced East Taylor watershed (ETW) in Colorado, consisting of two adjacent watersheds, one with, one without human influence by a reservoir. The existing National Water Model (NWM) in the watershed is used as hydrological model. The authors demonstrate that the GNN can almost fully emulate the NWM (Kling-Gupta efficiency in the order of 0.9). As the NWM itself performs relatively poorly when compared to observations (KGE in the order of 0.5), especially for high flows, the GNN is shown to further profit from fine-tuning against streamflow observations, and from data fusion, propagating error corrections along the river network connectivity, in case of forecasting. On this basis, the authors explore GNN setup and training variants depending on the physics-based model resolution and parameter structure.

**3. Evaluation**

Overall, this is a very well-written manuscript on a relevant topic. The introduction provides a very good overview on the topic, the goals of the study, the experimental design, the methods and results are very clearly explained, and all conclusions by the authors are supported by the results. Congratulations. I have only very comments:

- NWM model quality: Overall, the NWM does not perform very well in the ETW basin, and so does the pretrained and fine-tuned GNN (see Table 2). Can you explain in more detail why this is so? Also, I have seen many studies where LSTMs were trained on watershed data in the US, in daily time steps, for basins with snow influence, with higher performance. While I see the benefits of the author's approach to emulate high-resolution physics-based watershed models, I wonder what KGE in testing could be achieved by LSTMs simply trained on the available gauges. I suspect they would outperform both the NWM and the GNN. Again, this will not invalidate this study, as the goal is somewhat different. Nevertheless, training an LSTM at least on some or one of the gauges (e.g. at the basin outlet) and presenting the results would help to put the results of the study into perspective.
- Fig 6: In the figure caption, subplot c) is not explained

Yours sincerely,

Uwe Ehret

---

## Author Response (AR1)

Responses to reviewers' comments

[Original Comments in Red, Replies in **Black**]

Dear HESS Editor and Reviewers,

We appreciate your constructive comments, which help to improve our manuscript. In the revision, we added a new section to demonstrate the scalability of the GNN framework. We also added additional language to explain the fine-tuning issues as requested by Reviewer #2. The details are provided in the itemized rebuttal letter below.

Sincerely,

Alex Sun

**Reviewer 1 Comments**

This work presents a multistage, physics-guided, graph neural network (GNNs) approach for basin-scale river network learning and stream forecasting. This approach is computationally less demanding than vector-based river network models. I am a hydraulic engineer with some expertise in the modelling of river reaches, including under flood conditions. I have accepted the invitation to review this paper in the hope to be able to provide constructive and useful comments and suggestions to the authors, and in the hope to expand my own knowledge base.

Reply: Thanks for the summary and we really appreciate the careful review and comments from Reviewer 1.

I have enjoyed reading the paper and I have understood the concept of the method, which I find very interesting. I have the impression that the authors master the theme and have made a worthwhile contribution. But I have to admit that I am not sufficiently familiar with the topic to make an authoritative assessment of the quality and originality of the contribution.

Reply:  Thank you for the generous comments.

I hope that the following suggestions will be helpful to the authors:

The paper is very technical and probably not very appealing to non-experts in the field of neural network approaches. The authors may want to make an effort to make the paper more appealing to a broader readership.

Reply: We appreciate the comment. In preparing the manuscript, we endeavored to balance between the machine learning (ML) methodology contribution, scientific significance, and readability to elevate the manuscript to a publishable level. This is largely because a large body of peer-reviewed ML articles have already been published. We made multiple efforts to improve the readability of the manuscript, including:

- A general Introduction laying out the background and the need for spatiotemporal streamflow modeling.
- Figure 1 provides a high-level diagram describing the workflow, including data formulation, training steps, and data fusion. For most readers, this diagram should be sufficient to understand the content and contribution of this work.
- A multi-stage Result section that demonstrates the performance of each stage corresponding to Figure 1.

The methodology part includes several equations to describe how the graph neural net (GNN) works, which may lower readability. However, we argue the level of mathematics and jargons used are comparable to a typical ML paper involving long short-term memory network (LSTM) or convolutional LSTM (Conv-LSTM). For example, both GNN and LSTM use the latent variable extensively to transfer information. In the revision, we carefully revised the Abstract and the text, wherever appropriate, to further ease the readability.

The authors highlight that a major advantage of the graph-neural-network approach over a vector-based river model is the much lower computational demand. I suggest substantiating/quantifying this lower computational demand.

Reply:  Accept. The GNN framework presented in this manuscript embraces a "physics-based post-processing" approach, as described around L60-80 in the Introduction. Thus, there are two parts, the training part and the online operation part. The training part uses supervised learning, in which simulated data is required for the model to learn the behavior of the vector-based river model, where the prediction part uses past model outputs and meteorological forcing data to predict future streamflow. It is the latter part that we believe can significantly reduce the running time of vector based model. In the revision, we added the quantification of test time around L406, "The total running time is 5.4 sec wallclock time on the test data."

The model is only demonstrated for one relatively small snow-dominated watershed in the western US. Is this a sufficient basis for claiming general validity of the model in watersheds in other geographical settings.

Reply: Accept. During the revision, we tested the GNN framework on a much larger basin (20,000 km^2 vs. 1800 km^2) in the same Upper Colorado Basin. Results demonstrate that the performance of the GNN framework largely holds without much changes. We report the additional results in the newly added Section 6.

I suppose these suggestions amount for a moderate revision (something in between a minor and a major revision).

**Reviewer 2 Comments [**Uwe Ehret]

I have reviewed the aforementioned work. My conclusions and comments are as follows:

1. Scope

The article is well within the scope of HESS.

Reply: Thank you.

2. Summary

The authors propose a machine-learning (ML) approach to catchment hydrological modeling in daily resolution, consisting of graph neural networks (GNNs) pretrained on a distributed hydrological model (thus absorbing information about physical processes and subbasin connectivity as encoded in the model), fine-tuned on observed streamflow at available gauges (thus absorbing real-world information not captured by the hydrological model), and finally, if forecasting rather than simulation is the goal, a data-fusion approach (thus absorbing information of observations from the immediate past not captured by GNN modeling based on observed of forcing, but not streamflow). The authors demonstrate the approach at the example of the high alpine, snow-influenced East Taylor watershed (ETW) in Colorado, consisting of two adjacent watersheds, one with, one without human influence by a reservoir. The existing National Water Model (NWM) in the watershed is used as hydrological model. The authors demonstrate that the GNN can almost fully emulate the NWM (Kling-Gupta efficiency in the order of 0.9). As the NWM itself performs relatively poorly when compared to observations (KGE in the order of 0.5), especially for high flows, the GNN is shown to further profit from fine-tuning against streamflow observations, and from data fusion, propagating error corrections along the river network connectivity, in case of forecasting. On this basis, the authors explore GNN setup and training variants depending on the physics-based model resolution and parameter structure.

Reply: Thank you for the thorough and accurate summary.

3. Evaluation

Overall, this is a very well-written manuscript on a relevant topic. The introduction provides a very good overview on the topic, the goals of the study, the experimental design, the methods and results are very clearly explained, and all conclusions by the authors are supported by the results. Congratulations. I have only very comments:

  • NWM model quality: Overall, the NWM does not perform very well in the ETW basin, and so does the pretrained and fine-tuned GNN (see Table 2). Can you explain in more detail why this is so?

Reply: The deficiency of NWM has been noted by several works, especially in snow-dominated regions, e.g., Garousi-Nejad and Tarboton (2022). The pretraining is designed to mimic the NWM outputs as much as possible. So if NWM does not match observation well, the pretained model won't do much

better. The fine tuning is supposed to offer some calibration using training period gage data. In the revised manuscript, we offered additional explanations on lack of good performance due to fine tuning around L497.

*There can be multiple reasons. First, in this case, fine-tuning is restricted to a few nodes and the total effect is not as significant as when all nodes are tuned, such as in typical climate applications \citep{ham2019deep}. Second, the forcing data we used in driving the GWN is not accurate enough to allow the models to capture high flows. Nevertheless, the bias corrections resulting from the fine-tuning stage, especially in the phase of the time series, are important for the subsequent data fusion step.*

- Also, I have seen many studies where LSTMs were trained on watershed data in the US, in daily time steps, for basins with snow influence, with higher performance. While I see the benefits of the author's approach to emulate high-resolution physics-based watershed models, I wonder what KGE in testing could be achieved by LSTMs simply trained on the available gauges. I suspect they would outperform both the NWM and the GNN. Again, this will not invalidate this study, as the goal is somewhat different. Nevertheless, training an LSTM at least on some or one of the gauges (e.g. at the basin outlet) and presenting the results would help to put the results of the study into perspective.

Reply: We believe the Reviewer is referring to a series of data-driven LSTM papers published by Freddie Kratzert et al. in recent years on the CAMELS dataset. Indeed, their LSTM achieved state-of-the-art performance. However, using the same dataset and also training at daily steps, we showed recently that GNN and LSTM achieved comparable performance in terms of NSE cumulative distribution. In Figure 1, adapted from Figure 8 of our WRR paper, we showed that a pure data-driven GNN is slightly better than LSTM over Colorado watersheds.

So why didn't we take a data-driven approach here? As the Reviewer mentioned, our goal is different. We would like to be able to estimate streamflow at all ungauged locations (nodes) and so far process-based models seem to offer the most meaningful way. Of course, the price we pay is the biases introduced by model errors that are hard to eliminate using fine tuning alone. Nevertheless, after postprocessing, the results become much better. The other fact we'd like to point out is that CAMELS represents a carefully curated dataset that has eliminated anthropogenic effect (such as man-made reservoirs), while half of the watershed in ETW is affected by the Taylor Park reservoir. This human effect is hard to model for even data-driven ML models if the predictors (e.g., reservoir release time series) are not available.

[Figure]

*Figure 1. Comparison between GraphWavenet and LSTM over the CAMELS dataset. Gages falling in Colorado state are circled. Figure adapted from Figure 8 in Sun et al., 2021.*

- Fig 6: In the figure caption, subplot c) is not explained

Reply:  Changed.

**References:**

Garousi-Nejad, I., & Tarboton, D. G. (2022). A comparison of National Water Model retrospective analysis snow outputs at snow telemetry sites across the Western United States. Hydrological Processes, 36(1), e14469.

Sun, A. Y., Jiang, P., Mudunuru, M. K., & Chen, X. (2021). Explore Spatio-Temporal Learning of Large Sample Hydrology Using Graph Neural Networks. Water Resources Research, 57(12), e2021WR030394.